# The University and the Neighbourhood—Opportunities and Limits in Promoting Social Innovation: The Case of AuroraLAB in Turin (Italy)

**Francesca Bragaglia** 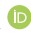

DIST—Interuniversity Department of Regional and Urban Studies and Planning, Politecnico di Torino, 10129 Turin, Italy; francesca.bragaglia@polito.it

**Abstract:** This paper deals with the relationship between university public engagement activities and local territories in promoting social innovation processes. In particular, this paper starts from the assumption that since social innovation has become a guiding concept of policies at various scales, opportunities for innovation, i.e., calls for tenders, funding, etc., have multiplied. However, universities should act as intermediary actors so that the bureaucratic and managerial complexities of accessing these opportunities do not risk cutting off the territories and/or the weakest actors. Starting from the experience of the AuroraLAB action–research laboratory of the Politecnico di Torino within the Tonite project financed with European Urban Innovative Actions funds, this article investigates the multiple roles that the university can play in supporting platform spaces for inclusive social innovation based on local needs. This article concludes by highlighting the multi-layered personality of the university in neighbourhoods and the perspectives for socially engaged research.

**Keywords:** social innovation; co-production; university public engagement; urban regeneration; neighbourhoods; Urban Innovative Actions; Turin

## 1. Introduction

Contemporary urban and territorial challenges are leading governments worldwide to rethink their governance models. Deep crises have so far characterised the new century: first, the global economic–financial crisis of 2008 and, more recently, the pandemic and energy crises. It is precisely in both these global and local crises—whose etymology comes from the Greek κρίσις meaning "to separate", "to choose", "to change"—that social innovations emerge as possible responses [1]. In a socio-cultural and administrative context in transition, where institutional actors are less and less able to support welfare policies and responses to complex, pressing, and place-based local needs, social innovation is indeed seen as an opportunity to do better with fewer public resources [2]. The literature on urban and regional studies has recognised how social innovation is now a pivotal concept in policy agendas at different scales of authority [3]. In particular, its success is due to the conceptual and normative breadth and consensus implications that characterise this concept [4,5].

According to the social innovation paradigm, new forms of collaboration between local socio-spatial actors are emerging and should be sustained and promoted [6]. These are new alliances between public, private, and third-sector actors and citizens to shape tailor-made responses to unmet needs [7,8] and for urban regeneration [9]. In this sense, social innovation is invoked by different actors as a fuse capable of triggering co-construction processes in policies and practices. The result is a framework in which the potential agents of social innovation multiply as well as the opportunities for innovation triggered by policies, programmes, calls, and collaborative governance tools. It is no coincidence that one of the main promoters of social innovation is the European Union, which, over the past two decades, has primarily supported the spread of a social innovation culture

among its member states [10] through a considerable funding scheme and various research projects and collaborative initiatives. Among these, a particularly noteworthy example is the Urban Innovative Actions (UIA) programme, which aims to test innovative solutions in complex urban contexts to address the issues discussed in the EU Urban Agenda launched in 2016 with the Amsterdam Pact. As Tricarico et al. [11] clarify, the UIA programme exemplifies a place-based territorial approach to social innovation by activating local knowledge of multi-sectoral networks of actors and bottom-up activation [12]. The UIA programme is the most emblematic recent experience through which the EU has mobilized to fund social innovation at the urban level. Therefore, it is essential to investigate the possibilities offered by UIA and the multiple actors that enable social innovation processes. Among these actors, we recognize the university, in its Third Mission action, as a central intermediary actor in social innovation processes. More precisely, this paper discusses how universities, playing different roles in local territories within public engagement initiatives, can be considered social innovation agents that support the co-production of situated answers to local problems in neighbourhoods. Although it is still a little-studied topic, both theoretically and empirically, the university's role in supporting local social innovation processes is crucial in the perspective of new governance models that enhance the know-how and expertise of a multiplicity of actors. Indeed, as Shiel et al. [13] (p. 126) clarify, "universities can foster the co-creation of community change by contributing research, technical expertise, human resources and emerging knowledge".

In this sense, this contribution intends to analyse the Turin (Italy) case of the ToNITE Project, financed precisely through UIA on "urban security", which mainly targeted the Aurora neighbourhood, a fragile urban area north of the city centre with a rich social capital. In this context, the AuroraLAB action-research lab, promoted by the Interuniversity Department of Regional and Urban Studies and Planning of the Politecnico di Torino within its Third Mission, has been active since 2018 to offer students opportunities for innovative teaching and support local stakeholders of the Aurora neighbourhood in micro-processes of urban regeneration and social innovation [14,15]. AuroraLAB, together with other local actors of the Aurora neighbourhood, won ToNITE-UIA funding in 2021 with a project named "Grandangolo—Dream Spaces for Safe Living". Starting from the case of AuroraLAB, this paper questions how university institutions can support social innovation processes [16] in neighbourhoods characterised by multiple socio-spatial complexities. As acknowledged in the literature on social innovation, it is often multi-problem neighbourhoods that are hotbeds of bottom-up social innovation [8,17,18], but for local communities and neighbourhood associations, it can be complex to intercept the resources offered by calls for tenders and other institutional opportunities offered by this new policy credo in social innovation. In investigating the case study of AuroraLAB in Turin from the perspective of social innovation, the opportunities offered by university action research in local territories are therefore brought into focus, in addition to the limitations of this type of experience. The empirical case discusses AuroraLAB's actions within the UIA initiative in the Turin context of Aurora, serving as an emblematic example of the university's role within a European program that promotes bottom-up social innovation in deprived urban contexts. This paper is the result of a long action–research experience and a self-reflective approach carried out during more than a year of activity in the Grandangolo project. This paper aims to expand theoretical and methodological reflection on the university's role in supporting social innovation processes and fostering the engagement of the most marginalized territories and social groups. Its purpose is to raise awareness among academics involved in public engagement initiatives regarding their specific role, the ethical, social, and political significance of their action research in deprived contexts, and the possible synergies with local stakeholders such as public officials, third-sector associations, activists, and the residents' representatives.

This paper is, therefore, composed of six sections. The first theoretical section discusses social innovation from an ecosystem perspective, identifying its drivers and agents and how the university's public engagement can be a piece of this ecosystem. Drawing from previous

work regarding the university's place in social innovation and its intermediary positioning, the methodological section identifies the university's specific roles in supporting these processes, which are then investigated in the case of AuroraLAB in Turin. Then, the discussion section discusses the limits and opportunities of university engagement in supporting social innovation at a neighbourhood level. Finally, the conclusions clarify the role universities can play in the local territories in the framework of governance increasingly based on proximity and multi-agent knowledge.

## 2. Social Innovation Ecosystems: What Is Changing and the Place for University Public Engagement

Although the concept of social innovation has a long tradition in various fields, it is only in the last 15 years that it has become a steering concept for supranational bodies and governments at multiple scales. In one of the most widely accepted definitions in urban studies [8], social innovation is defined as a process characterised by three basic aspects:

1. The satisfaction of unmet human needs and the consequent improvement in the living conditions of local communities;
2. The change in socio-spatial relations between the actors involved in the process;
3. The empowerment of local communities and, as a consequence, better access to collective resources.

It is clear from this definition that the humus of social innovation is the local communities themselves (third-sector actors and self-organised citizens), which identify local needs and possible place-based solutions. As recognised by Lazzarini and Pacchi [19], nowadays, this grassroots agency is increasingly hybridised with other types of social innovation agents. In order to occur and sustain, social innovation needs policies, resources, and enabling platform spaces [20]. Not by chance, the idea of "doing together" has become a steering principle of contemporary urban governance and spatial planning [21] to promote new mechanisms of shared responsibilities and collective understanding [22]. In this sense, several authors have started to look at social innovation from an ecosystem perspective, which sees the local milieu—composed, therefore, of elements such as space, actors, resources, policies, and their interrelation—as a crucial element for enacting social innovation experiences [11,23,24]. As Tricarico et al. [11] stated, "SI-based policies have been discussed as possible solutions to cope with the impositions related to the issue of proximity as a driving factor to recalibrate the spatial reorganization of services and to the management of social dynamics" (p. 2, emphasis added). We are thus witnessing a renaissance of "the local" [25], where the neighbourhood scale becomes the target of policy experimentations based on co-production with public administration [26–28]. It is at the neighbourhood scale that a strong interdependence emerges between the place—understood as a specific local context that expresses peculiar characteristics and needs—and the people, i.e., the subjects that inhabit that place and, in some cases, take action—together with other agents—to develop social innovation processes [29] at the local scale, where different forms of knowledge can encounter each other and synergise to solve context-specific issues. It is no coincidence that Sandercock [30] calls for an "epistemology of multiplicity" that draws on a variety of actors and their respective knowledge. This invites a move from a monolithic view of knowledge as essentially expert toward a more multi-layered and enlarged conceptualisation of knowledge, which also takes into account non-expert expertise and the contribution it can provide to urban wellbeing [31].

Moreover, the emergence of social innovation as a magic concept for policymakers [5] has multiplied "innovation opportunities" based on co-production. These, although configured as a response to unmet local issues, are stimulated by the huge resources offered through calls for proposals promoted, among others, by the European Union, national governments, and bank foundations. The resources allocated can be crucial in enabling local social capital to sustain itself [6,32] and generate new social innovations [19]. However, access to funding is often demanding as it requires responding to complex and highly competitive calls. In this sense, the most fragile territories, although often rich in social

capital, may have more trouble intercepting the economic resources launched by these policy opportunities, especially regarding informal neighbourhood groups. If, on the one hand, institutionalised social innovation and related co-production experiences try to broaden the arenas of governance and shared projects, on the other hand, technical and bureaucratic complexities may constitute a barrier for less structured social actors and/or more deprived territories.

A growing part of the literature is not surprisingly focusing on so-called "intermediary actors", who emerge as a result of the new opportunities offered by socially innovative initiatives based on the idea of co-production between public institutions and local communities [33,34]. Intermediary actors support these collaborative processes by providing specific expertise and connecting the actors involved. Intermediaries are a composite and multifaceted universe of subjects who act for profit or not [35]. Among the latter, universities are recognised as a possible intermediary actor, "which can play an expert but 'third' role, taking part in processes with a propensity for self-reflection and cultural autonomy" [36] (p. 26). Universities can help coalesce, organise, and facilitate networks of social innovation agents around common goals and intercept institutional resources and funding that enable adequate support for social innovation processes [15] (Figure 1). In this sense, university institutions not only reflect on the topic of social innovation from a theoretical point of view but are also more and more implicated in it from a practical point of view as part of their "third mission" [37]. The university's third mission includes activities of a different nature and has so far been understood more in terms of technology transfer to territories, while only more recently is the idea of an "engaged university" [38] towards territories gaining momentum [39,40].

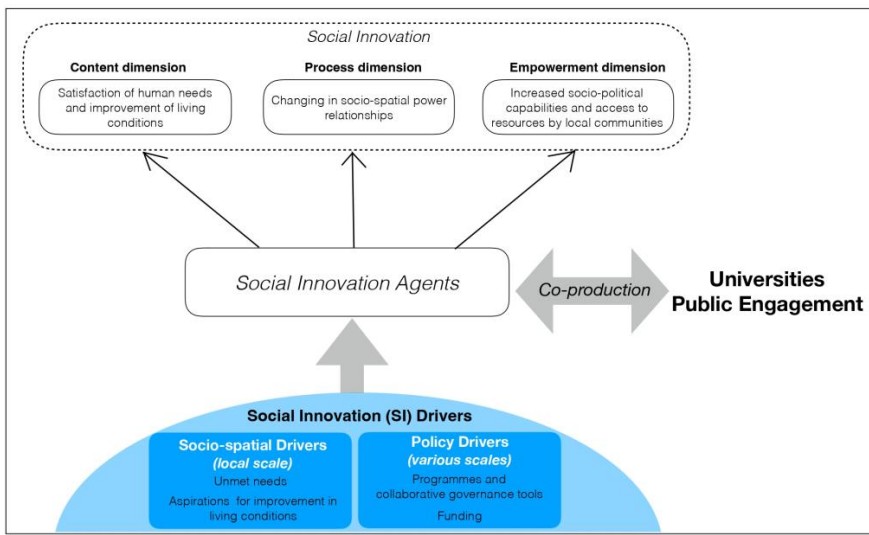

**Figure 1.** Social innovation drivers and agent; author's processing.

Assuming, therefore, that the university can be an essential element in supporting territorialised processes of social innovation that are sustained through institutional funding resources—as, for example, in the emblematic case of the European UIA programme (now relaunched under the new EUI-IA programme)—it becomes crucial to understand better how this happens using case study analysis.

## 3. A Methodological Note on Case Study Analysis

As seen in the previous section, within social innovation ecosystems, a growing body of literature is investigating the specific role that university institutions can play by providing support to local territories and their social innovation agents. Drawing on the literature that has studied the role of the university in supporting social innovation processes and

the related co-production of solutions at the local scale [34,41,42], it is possible to recognise six specific roles that the university in its public engagement can assume:

(i)    "Process activator", namely, a subject able to intercept external opportunities useful to activate social innovation processes.

(ii)   "Knowledge(s) collector", namely, a subject able to put together and systematise different forms of knowledge.

(iii)  "Research provider", namely, a subject that supports the social innovation process thanks to its technical skills.

(iv)  "Mediator", namely, a subject facilitating the connections and the interaction between all the actors involved and mediating the processes of involvement of the other actors involved in the social innovation process.

(v)   "Operative", namely, a subject able to collaborate in the practical activities of the process.

(vi)  "Knowledge broker", namely, a subject who disseminates the knowledge produced through wider networks of actors and to resources not directly included, contributing to uptaking good practices.

Within the social innovation process, the university can play a specific role among those listed or all of them, depending on the process's needs, stages, or typology. In an ecosystem logic, the university interacts with other institutional and non-institutional actors and supports local neighbourhoods while having its own recognised and autonomous agency. Starting from this cognitive framework, this article discusses the case of the AuroraLAB-Politecnico di Torino action–research laboratory that has been active since 2018 in a deprived neighbourhood with a rich social capital and its agency in supporting social innovation and producing usable knowledge for the Aurora community. The research approach was based on field research and, therefore, a continuous and prolonged presence in the Aurora neighbourhood of the action–research group during six years of activity. The resulting self-reflection focuses on the effects of the university's presence in this neighbourhood and its multiple roles in accompanying the territorial realities involved in an articulated and multi-faceted project such as that of the Grandangolo project within the broader ToNITE-UIA project (Figure 2).

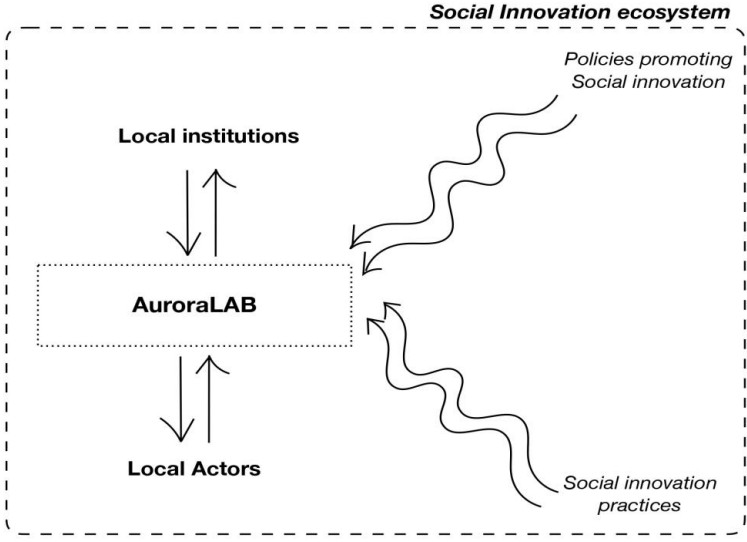

**Figure 2.** AuroraLAB as an intermediary actor in Aurora's social innovation ecosystem; author's processing.

AuroraLAB's involvement in the Grandangolo project used various qualitative and quantitative research tools to assess the results obtained. On the one hand, participant observation was conducted throughout all project phases (ex ante, in itinere, and ex post). On the other hand, thirty interviews with local stakeholders and an "impact assessment framework" facilitated the collection and analysis of actions carried out in the neighbourhood within the Grandangolo project and the impact on the beneficiaries. The latter was divided

into three impact areas with specific indicators: (1). involvement and active participation, (2). Knowledge, and (3). liveability of public space. The different data collection methods facilitated shedding light on AuroraLAB's positioning vis-à-vis other partners at different project stages.

## 4. AuroraLAB and the Case of the Grandangolo Project in Turin

### 4.1. AuroraLAB's Rooting in the Local Aurora Neighbourhood Socio-Spatial Ecosystem: An Incremental Process

Aurora is a former industrial neighbourhood—whose fabric is marked by numerous urban voids—close to the city centre, characterised by deep socio-spatial fractures (Figure 3). It is one of the poorest neighbourhoods in Turin. However, in recent years, it has undergone numerous urban transformations, leading to progressive gentrification of parts of the neighbourhood. Concerning the social fabric, Aurora is characterised by a low level of education and high unemployment. Moreover, it is one of the city's youngest areas, with an under-15 population steadily increasing in recent years and a profoundly multicultural area. A further element that marks the neighbourhood is linked to urban insecurity, sometimes actual, but more often in the perception of the individuals, and which over time has constructed the stigma that characterises Aurora. It is, therefore, a complex area but with a robust and resilient social capital operating in the territory. In this sense, it is a fertile ground for social innovation processes.

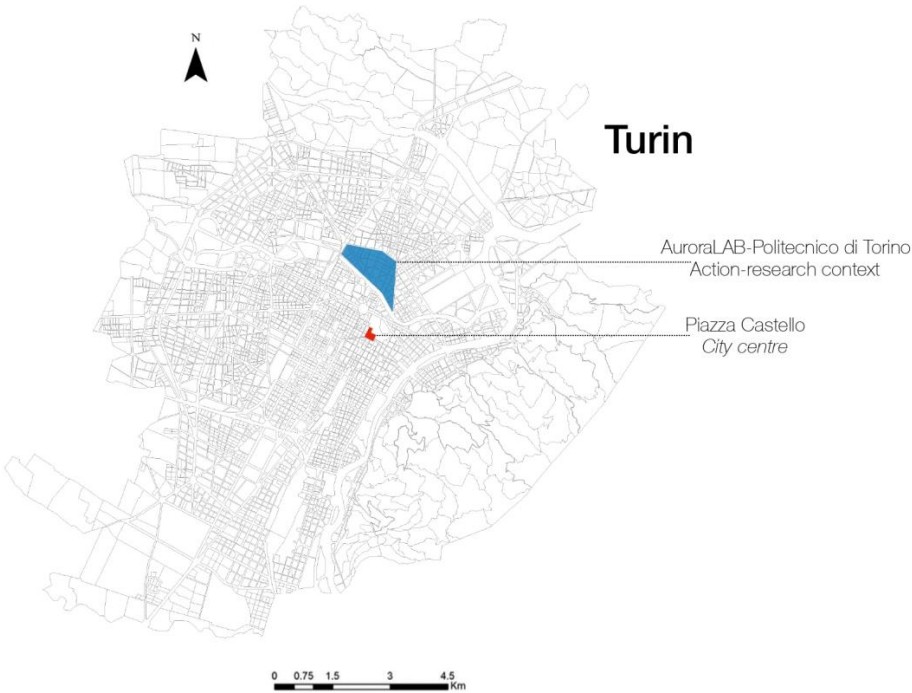

**Figure 3.** The Aurora neighbourhood in the Turin context; author's processing.

In this multi-faceted neighbourhood, the AuroraLAB action–research laboratory was launched in 2018. It constitutes a pilot laboratory in the Turin context, involving researchers with different academic profiles (including urban planners, geographers, sociologists, and economists) united by a triple challenge: teaching, research, and public engagement [21]. In this sense, as recognised by Caruso et al. [14], AuroraLAB poses several orders of challenges. Regarding didactics, the challenge is introducing forms of teaching outside the classroom, bringing students into direct contact with the real and complex issues that the territories demand. Regarding research, the challenge is to build knowledge with an interdisciplinary and field-based approach through listening and dialogue with local act The image has been modified as requestedors and the Aurora neighbourhood. Lastly, perhaps AuroraLAB's most ambitious challenge is to give something back to the territories

investigated by bringing the university closer to the neighbourhood, especially in a context such as that of Aurora, which often has latent resources that need to be valorised, put into a system, and supported in their actions. In this sense, AuroraLAB's participation in ToNITE UIA is only the latest step in a long and complex process of progressive territorial rooting of this research–action laboratory of the Politecnico di Torino in the socio-spatial fabric of the Aurora district. The progressive acquisition of recognisability and legitimacy to act in the neighbourhood has been a long process, as AuroraLAB was an "outsider" regarding Aurora's socio-spatial ecosystem.

For this reason, in an initial phase of activity in the area, AuroraLAB limited its activities to research through the construction of two reports on the neighbourhood and a significant campaign of interviews with privileged local actors of the area (local associations, neighbourhood committees, advocacy groups, etc.), to get in touch with them and establish networks. AuroraLAB's process of progressive territorial rooting has also passed through its participation in the "Rete Coordinamento Aurora" (Aurora Coordination Network), a bottom-up mutualism network created to respond to the COVID crisis in the area, which is still active nowadays and involves about 40 different local realities of the neighbourhood with a horizontal governance approach. AuroraLAB has, therefore, worked to carry out meaningful research for and with the neighbourhood and to build its specific identity as a trustful actor of Aurora until it took the opportunity offered by the Turin Municipality's call for tenders on the ToNITE-UIA project in 2020.

*4.2. The Grandangolo Project*

Grandagolo is one of 19 projects in the Aurora and Vanchiglia neighbourhoods of Turin selected by the Municipality of Turin as part of the TONITE project with Urban Innovative Action funds. The objective was to improve the perception of urban security not through policies of securitization of urban public space (e.g., installation of cameras or police garrisons), which have demonstrated several limitations, but through collaborative practices based on community empowerment and the active participation of local inhabitants and stakeholders. Public space is, in fact, both a material and immaterial space, a relational aggregate where formal and informal, inclusive or exclusive activities are articulated. The TONITE project explicitly aimed to enhance the safety of public spaces in the neighbourhoods of Aurora and Vanchiglia during evening and night-time hours, ensuring community-based urban security. The selected projects, therefore, had to address this fundamental demand through solutions in line with UIA's guiding principles (innovation, participation, quality, measurement, and transferability).

The selected projects were 80% financed by Tonite-UIA funds, while the remaining 20% was co-financed by the partners of each project. Grandagolo started in September 2021 and ended in November 2022. It was the project with the largest partner network among all 19 funded projects. Indeed, it saw the collaboration of nine very different local actors coordinated by AuroraLAB—Politecnico di Torino: four third-sector associations working on various topics (art and culture, sport, cultural mediation, and education), a primary school, two neighbourhood committees, and the Luigi Bobbio Research Centre of the Università di Torino (Figure 4).

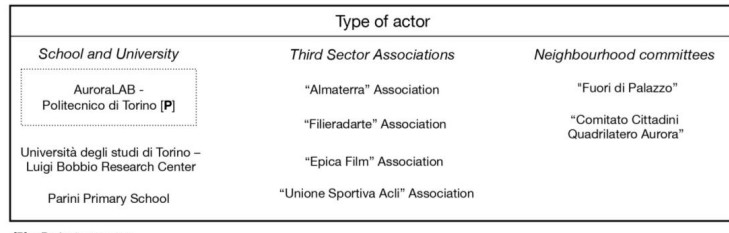

**Figure 4.** Social innovation agents within the Grandangolo Project; author's processing.

These social and institutional entities, so different from each other, were united in a "community of practice" [43]. Following Moroni and Tricarico [44], the concept of community is understood in non-ideological terms as individuals who share a common goal and are part of an organisational system that regulates their actions. In the case of Grandangolo, the common goal was to promote a participatory and inclusive security model in the neighbourhood, particularly during the evening hours, through cultural activities of public space praesidium and tactical urbanism interventions in specific areas of the neighbourhood. Leveraging the value of encounter and social contact generated in public spaces, Grandangolo has promoted forms of re-appropriation of those spaces with multiple subjects (children, foreign women, and families living in the Aurora neighbourhood). In this sense, the project focused on activities such as performances, dance and theatre classes, workshops, cultural and language mediation activities, Neighbours' Day, etc.

The idea was based on the recognition of social capital as the leading resource of the neighbourhood, which can be capitalised and structured through the development of a project capable of activating tailor-made collective initiatives. The Grandangolo project, therefore, straddled the social dimension—i.e., the sharing of ideas, the facilitation of reciprocal knowledge among the different communities living in Aurora, and the construction of relations of trust among the neighbourhood's inhabitants—and the spatial dimension (re-appropriation of public spaces, construction of a collective identity of places, self-produced tactical urbanism solutions). The aim was to generate social innovation starting from an unmet need, that of security, through strengthening the socio-spatial relationships between the actors involved and empowering local people. The project is composed of small actions whose value does not lie solely in the final output but rather in the process that led to that output, and which aims to create shared value, social resilience, and inclusion [45]. In order to discuss the place of the university in supporting these types of social innovation processes that link up with institutional opportunities offered by calls for proposals (as in the case of UIA), it is essential to look at the specific roles that AuroraLAB assumed in the different project phases (Figure 5).

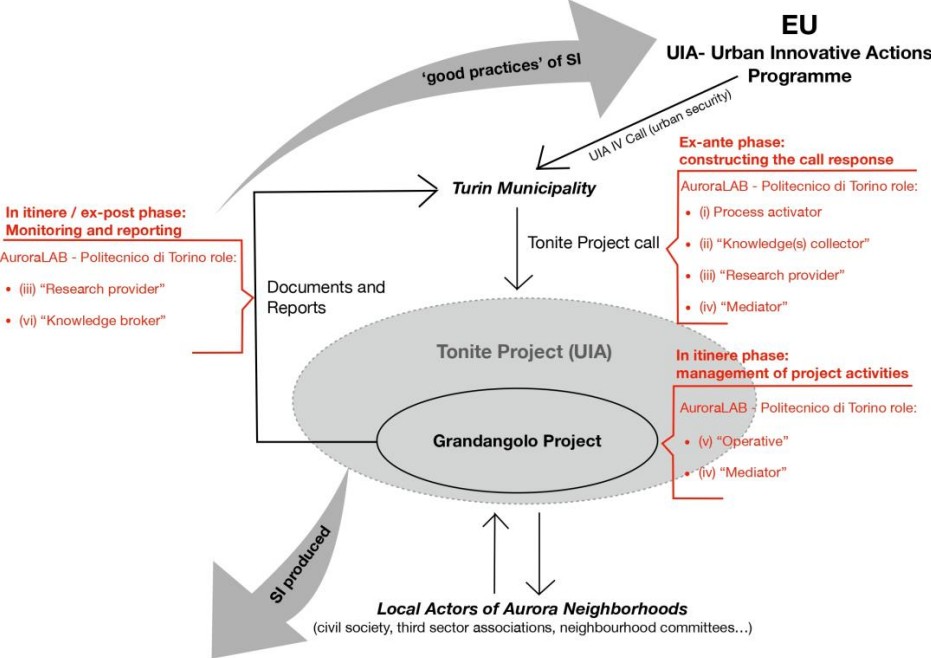

**Figure 5.** AuroraLAB's roles in the different phases of the Grandangolo project.

4.2.1. Ex Ante Phase

The ex ante phase was characterised by the project partnership's construction and the project proposal. This was a crucial phase for obtaining funding in which AuroraLAB played several roles. First of all, it played an activator role (i) by identifying the opportunity

offered by the Tonite-UIA call for proposals and acting as a collector of the different types of knowledge (ii) that the other eight project partners were providing (local knowledge, technical knowledge, etc.). It was therefore necessary to mediate and bridge (iv) between all the actors and their different know-how to create a common project. In response to the call and drafting of the project proposal, AuroraLAB's role was then essentially that of a "research provider" (iii), thanks to the expertise of the researchers of the Politecnico di Torino in European and national calls for proposals and knowledge of the local context acquired in previous years of action–research in the neighbourhood.

### 4.2.2. In Itinere Phase

Once the Grandangolo project was awarded funding, it moved into the operational phase of running the project activities over a period of one year and two months (from September 2021 to November 2022). Overall, Grandangolo implemented 67 initiatives in the neighbourhood in the evening or pre-evening hours and involved approximately 3800 people. The initiatives were multiple and site-specific, focusing primarily on the target group of school children. In the neighbourhood, there is, in fact, a high percentage of foreign children from different cultures. In the primary school included in the Grandangolo project, this percentage is over 75 per cent. Young people from deprived backgrounds often find it difficult to express their voices [46], so giving them an active role in the participatory transformation of the neighbourhood's public space was paramount.

Moreover, each of the Grandangolo project partners made their specific knowledge and expertise available to the project. Throughout this phase of the Grandangolo project, AuroraLAB took on various roles to support the local social innovation ecosystem. In this phase, AuroraLAB's mediation (iv)—as project leader—between the different partners was important in organising and scheduling the different actions to build synergic solutions. AuroraLAB also acted as a collaborative bridge between the representatives of local institutions and technicians of the Turin Municipality involved in the Tonite project and Aurora's local actors, in order to follow the accompanying and monitoring activities included in the very stringent requirements of the Municipality of Turin and the European Union. Within this framework, an impact assessment framework was built—together with the Luigi Bobbio Research Centre of the University of Turin, the other university-based project partner of Grandangolo—which allowed the evaluation of the actions carried out based on some specific indicators related to three impact areas, i.e., the ability of the actions to foster the involvement of the neighbourhood and the active participation of all actors; the construction of new knowledge; and the improvement in the liveability of the public space.

In addition to its role as mediator, AuroraLAB's role in this phase of the process was also operational (v) in carrying out its activities and providing support to the other project partners. In fact, AuroraLAB coordinated a series of actions in the neighbourhood's public space, particularly those related to a tactical urbanism intervention along the pavement in front of the project partner primary school, a co-design workshop that saw the students of design and planning courses at the Politecnico di Torino and the primary school children working together. The primary school children brought their experiential knowledge as users of the school and the public spaces around it, while the Politecnico di Torino students brought their technical knowledge of public space planning and design according to the needs and preferences indicated by the students.

### 4.2.3. Ex Post Phase

Once the project was concluded, the ex post phase was devoted to the elaboration of a series of rather complex technical documents on the activities conducted and the evaluation of their impact with qualitative and quantitative to share "good practices", and also for the project's economic reporting of the expenses sustained by the partners. In this phase, AuroraLAB's roles were mainly research provider (iii) and knowledge broker (vi). On the one hand, the project phase was characterised by the management of the project's bureaucracy to obtain funds supported by the work of the technical–administrative apparatus of the

Politecnico di Torino in charge of the economic management of the Grandangolo project. On the other hand, this phase was characterised by the need to communicate the innovation produced "on the ground" to different audiences and disseminate the good practices from UIA through reports and scientific publications. In both respects, AuroraLAB—also assisted by the specific work of the Università di Torino on monitoring and evaluating the project's impact—has been a key player. In particular, the monitoring data revealed how the Grandangolo project was able to increase the empowerment of local community actors through forms of co-design and social impact initiatives in the public space, an increase in the sense of belonging to the neighbourhood and its resources in terms of public space, and an improvement in the use of services in these spaces. AuroraLAB also promoted a multi-faceted narrative of the neighbourhood and attempted to unhinge the mainstream and flattening one, which tends to respond to complex issues with a socio-spatial bipolarity between "good neighbourhoods" and neighbourhoods, such as Aurora, that instead seems to concentrate on all the urban malaise, creating a prejudice based on peoples' postcode address [47]. The construction of a different narrative for the Aurora neighbourhood was also made possible thanks to the communication of the activities carried out to local and national newspapers. As a result, these newspapers reported on the Grandangolo project multiple times through their web and print media channels. Finally, at this stage of the project, the partners, coordinated by AuroraLAB, reflected on the legacy of the Grandangolo project. While the participatory urban security model has proved its effectiveness in increasing public space usage and care as well as building new collaborations among residents and local stakeholders, there is an issue related to the temporary nature of most of the actions taken. In contrast to hard security measures such as the installation of cameras, participatory security requires triggering new behaviours and ways of using and taking care of space. To ensure a visible legacy of the socio-spatial relations built during the Grandangolo project, AuroraLAB has coordinated the application project for the activation of a collaboration pact between the Parini Primary School and the surrounding public space under the Regulations for the Governance of Urban Common Goods adopted by the City of Turin in 2020.

## 5. Discussion: Roles and Significance of the University in Neighbourhoods

The 372 million allocated by the EU through the European Regional Development Fund in the UIA programme's five years of activity has been a significant support for social innovation processes at the urban scale throughout Europe. However, the bureaucratic complexities and technical skills required to capture and manage funding can hinder less organised local groups. In this sense, the university can provide a crucial agency to support the most fragile actors and to mediate between different actors who came together to intercept the public resources offered by programmes such as UIA and respond to context-specific problems. The network of local partners of the Grandangolo project coordinated by AuroraLAB is an emblematic example, as it brought together actors with very different vocations, knowledge, and organisational forms around the common goal of building participatory neighbourhood security at night and in the evenings, thus responding to an unsatisfied need. In particular, the two neighbourhood committees included in the Grandangolo project, being informal and somewhat fragile entities—albeit bearers of essential experiential and context-specific knowledge—would hardly have been able to benefit from the resources offered by the Tonite-UIA project if not for the support of AuroraLAB and the partnership of actors included in Grandangolo. A similar discourse applies to the neighbourhood primary school, an institution very active in the Aurora context but without a specific propensity to participate in calls for proposals, especially those as articulate and complex as the Tonite-UIA case. Consequently, AuroraLAB's mediation helped to make it possible for the school to participate. The latter contributed significantly to the project actions without, however, having to personally follow some of the more technical and complex passages of the call, which AuroraLAB managed. The multiple roles that AuroraLAB has been able to play in the Grandangolo project clarify the multi-layered professionality

that university institutions can provide to neighbourhoods. Indeed, AuroraLAB's experience in the Grandangolo project (Tonite-UIA) highlights the multiple roles that university institutions' public engagement activities in local territories can play in supporting social innovation processes. As seen in the previous section, intermediation of the university in the social innovation processes of territories that engage exogenous resources to sustain themselves, such as the case of Tonite-UIA can be divided into six specific roles experienced in the different project phases: (i) "Process activator", (ii) "Knowledge(s) collector", (iii) "Research provider", (iv) "Mediator", (v) "Operative", and (vi) "Knowledge broker".

Public engagement activities allow new relations with local stakeholders to be triggered with a view to action-oriented research and strengthen the university's role as a social actor and agent of social innovation processes. In this sense, the two-way relationship between universities and local territories can foster a dynamic and reflexive process in which universities may bring research and expertise related to emergent discussion within partnerships aimed at social innovation and can receive social legitimation and different knowledge inputs for new research [42]. In this sense, it must be acknowledged that the positioning of university institutions in the territories, especially in the most fragile ones, is a delicate process that requires time and the building of a relationship of trust with the actors in the territory. It is no coincidence that for the first two years of activity in the neighbourhood, AuroraLAB worked almost exclusively on building relationships with local stakeholders. As Cognetti [36] (p. 31) acknowledges, universities in the territories are usually perceived "as temporary and unstable presences, often 'taking' from the context without offering anything in return". It is, therefore, essential to deconstruct these beliefs with an ongoing and non-episodic presence supporting local territories and their potential for social innovation. In the Italian context, an interesting example to strive for is the Milan "Off-Campus" model developed by the Politecnico di Milano. These are stable experiences of the university's public engagement in deprived neighbourhoods—by now, four campuses are active and are present in the neighbourhood with a physical location, starting with the first experiment set up in the deprived neighbourhood of San Siro in 2013 [48]—which are at the same time replicable and context-specific. The interest in experiences such as that of "Off Campus" lies in the scalability of the operation and its ability to design innovation opportunities through a platform space where institutions, local actors of various kinds, inhabitants, and engaged researchers can pool different competencies to find answers to unmet needs in a win–win exchange.

However, it is difficult to assume that the relationship between universities and territories can be free of actual or apparent contradictions. It is important to consider possible power inequalities between academic institutions and local communities. While the social role of universities is recognised, on the one hand, the university's action in the territories is very often oriented towards placemaking dynamics, on the other hand. Indeed, universities can be direct and indirect promoters of the real estate development of neighbourhoods, and the impacts of these transformations are not always "just". An extensive portion of the academic literature is now focusing precisely on the role of the university in the transformation processes of certain urban areas, both concerning housing—i.e., rent growth—and concerning the complementary offers, such as leisure venues and premises designed for the student target and going so far as to coin the concept of studentification [49,50]. The case of AuroraLAB has shown the complexity of "entering" a neighbourhood such as Aurora, where the effects of studentification are now tangible due to its proximity to some university campuses and will be so especially in the following years, and the resulting impact on the more fragile population of the neighbourhood [51,52]. For this reason, some neighbourhood groups mistrusted AuroraLAB and its local action as representatives of the Politecnico di Torino. It is, therefore, important to be aware of the potential conflict that university action in neighbourhoods can generate. In this sense, it is once again crucial that the university shapes its reputation within neighbourhoods and establishes trusting relationships with local actors, acting not as a demiurge but as an enabler of social innovation processes, placing itself at the service of local actors. In other words, the university

should avoid adopting a paternalistic attitude towards the community, assuming the role of a saviour. The key to successful public engagement is to establish fair partnerships and acknowledge the local knowledge and experience gained through the daily lives of people in the neighbourhoods.

Finally, a final order of issues concerns the support of these public engagement initiatives by the respective university institutions. At least in the Italian context, the support given to this type of social action towards deprived local territories still needs to be improved in the face of a greater emphasis on a third mission understood primarily as technology transfer. Consequently, continuous financial support by university institutions would be helpful to ensure greater continuity of action. Without an ongoing source of funding that can only be provided by the university institutions themselves for this type of public engagement initiative undertaken by groups of engaged researchers, the risk is that these initiatives will be forced to constantly chase the opportunities offered by competitions whose outcome is very uncertain. This is all the more true if we consider that many of these experiences rely currently, above all, on the time and expertise of engaged researchers, and in an academic world increasingly marked by a "publish or perish" culture, action–research at the service of fragile local stakeholders risks being seen by many as an unstrategic choice because it is very time-demanding.

## 6. Conclusions

Contemporary public policies and practices appear to be increasingly characterised by a collaborative dimension to address multiple socio-spatial needs. In this context, social innovation has become a guiding concept in policymaking, leveraging civil society to find new solutions to various unsolved urban issues. This also requires the questioning of conventional knowledge production processes. Starting from a theoretical reconstruction of the new collaborative paradigm, this article highlighted how, on the one hand, the opening of governance arenas is a great opportunity for non-expert actors, while on the other hand, it highlighted the barriers for fragile subjects or the most marginal territories. There is, therefore, a risk of selective access to calls and funding that promote social innovation at different policy scales.

Parallel to these discourses on collaborative governance and social innovation, the academic literature has, not surprisingly, also highlighted the emergence of several types of intermediary actors capable of accompanying these processes [31,33,35]. If, on the one hand, this is opening up a new age of "consultocracy" for the private sector in accompanying collaborative processes, on the other hand, this also raises new questions about the possible positioning of university institutions within these ecosystems of actors revolving around social innovation and related policies and practices. This paper thus investigates the role academic institutions can play, recognising a robust social vocation within the framework of their Third Mission. In the Italian context, as in other countries, universities are increasingly exploring how they can support local social innovation processes by applying their expert skills in the service of local territories, especially the most fragile ones that risk being excluded, to (co-)produce action-oriented research and intercept the institutional opportunities offered by the new policy credo based on social innovation and the related collaboration between institutions and civil society actors. In this respect, this article clarified how the university can be an intermediary actor in these processes to support those who find it more challenging to engage in formal opportunities for social innovation and construct answers to unmet local needs.

Using the case of the AuroraLAB—Politecnico di Torino laboratory in the Aurora district of Turin, this article investigated this topic that is still relatively little dealt with in the literature even if it is expected to become progressively more relevant as urban governance is increasingly characterised by models that hybridise different subjects and types of knowledge and expertise. Indeed, building inclusive societies and facing contemporary challenges in the current scenario requires deploying place-based and tailor-made solutions based on the proximity and engagement of new actors. The multiple roles that the univer-

sity can play characterise its multi-layered professionality that can be put at the service of deprived territories, as well as contribute to the construction of "enabling spaces" [36] in urban peripheries. On the one hand, therefore, the presence of the university's public engagement experiences in neighbourhoods can be a multiplier of opportunities for territories. On the other hand, constructing a long path of mutual trust is important so that its engagement is seen as something other than a sort of "social-washing" in the face of the university's involvement—direct or indirect—in somehow exclusionary placemaking processes. In summary, university public engagement in fragile neighbourhoods can be vital for fostering social innovation. However, for it to be truly effective, it must be grounded in a critical understanding of power dynamics and a genuine desire to establish collaborative and authentic relationships with local residents.

Finally, given the increasingly prominent role of university public engagement in addressing profound socio-spatial inequalities and supporting socially innovative processes in various territories, future research on this topic is expected to be enriched with a broader empirical analysis that effectively explores various forms of synergy between universities and their respective communities. Additionally, an international comparative analysis would facilitate the comparison of public engagement approaches among university institutions worldwide. This would help identify common challenges, "good practices", and cultural differences that may impact the effectiveness of these approaches.

**Funding:** The Grandangolo project was financed within the framework of ToNite, a project activated by the City of Turin and co-financed under the fourth call of the European UIA-Urban Innovative Actions programme on Urban Security.

**Institutional Review Board Statement:** Not applicable.

**Informed Consent Statement:** Inform consent was obtained from all subjects involved in this study.

**Data Availability Statement:** Data are contained within this article.

**Acknowledgments:** The author of this paper would like to thank all the people involved in AuroraLAB and its partners in the Grandangolo project (ToNite UIA). Many thanks also to the guest editors of the special issue for this interesting opportunity for collective reflection.

**Conflicts of Interest:** The author declares no conflicts of interest.

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
