# Peer review of "The University and the Neighbourhood—Opportunities and Limits in Promoting Social Innovation: The Case of AuroraLAB in Turin (Italy)"

_sustainability, doi:10.3390/su16020829_

Round 1
Reviewer 1 Report
Comments and Suggestions for Authors
The article concerns an interesting issue: the activation of local communities and the role of the university in this process. However, the article does not contain specific effects of the mentioned project. It is only mentioned that the purpose was to ensure safety in the evenings. But isn't that what the police or city guards are for? What specific effects does such a project bring, what tasks are carried out, and who carries out these tasks? If we are talking about Italy, it has a rich tradition of cooperatives, when people came together to meet their economic and social needs. Or the activities of Catholic priests like Father John Bosko - also a grassroots initiative activating young people. If such projects must be financed from EU funds, what do they involve? Please specify the subject and subjective scope of the article.
Best regards
Reviewer 2 Report
Comments and Suggestions for Authors
Without a shadow of doubt, the social dimension of innovation is growing due to multiple challenges we all face. In this way, it is necessary to put emphasis on the universities and, in particular, on their role in social and sustainable development.
Universities can play a significant role in addressing social issues. They should go beyond their traditional missions, taking an active role by working with their local communities. I consider the topic of this article interesting from an academic perspective. In other words, I consider the article relevant.
In my opinion, the article starts with a correct introduction to the topic and defines widely the concept of social innovation -and its three basic aspects-. Starting from the case of the AuroraLAB-Politecnico di Torino laboratory, the aim of the article is clear, from the perspective of how the university’s public engagement actions can help to support social innovation processes.
Literature review, references, structure and methodology are also appropriate. Figures [5] are also right. Moreover, it may be possible to continue with subsequent lines of work.
Finally, I would make two recommendations: 1. To further elaborate the conclusions and 2. To clearly define who this article is addressed to (from a stakeholder perspective).
Reviewer 3 Report
Comments and Suggestions for Authors
The article is devoted to a current topic - the role of universities in supporting the processes of social innovation at the local level. The authors analyze the multiple roles that the university can play in supporting social innovation at the district level. It is shown that the university acts as a knowledge collector, researcher, mediator, and more.
The article contributes to the understanding of synergies between universities and territories in the field of social innovation. Further research could be devoted to quantifying the impact of universities on the development of innovation in urban areas.
The article does not clearly justify the choice of the specific case of AuroraLAB. Why was this project chosen?
There is little critical analysis of the limitations of the role of universities in territorial innovation.
The research methodology is described schematically; details of data collection and analysis for AuroraLAB are not provided.
More extensive empirical evidence is required.
